# A Term Extraction Approach to Expert Finding on the COVID-19 Open Research Dataset

## Abstract

The wide-scale impact of the COVID-19 crisis has brought to the forefront the need to identify experts with different scientific backgrounds. But controlled vocabularies and manually annotated publications lag behind when searching for recently emerging topics and well established scientometric approaches are notoriously difficult to compare across scientific areas. In this work we investigate a term extraction approach that automatically identifies expertise topics and builds expert profiles from scientific publications. We analyse domain-specific multi-word expressions related to COVID-19 and other related coronaviruses, and we discuss expert finding for five relevant ERC (European Research Council) life science areas.

## 1 Introduction

The COVID-19 crisis has focused the minds of scientists across the world on possible contributions in understanding the virus and the pandemic that it caused, and mitigating direct or indirect effects on people's lives. The wide scale impact of this crisis calls for diverse expertise from different fields of science, and brings an increased need for interdisciplinary collaboration and sharing of Covid-19-related knowledge to enable truly collective intelligence. Typically, researchers rely on metrics of publication and citation impact to evaluate the level of expertise of their peers, but these metrics are not as reliable nor directly comparable across scientific fields and bibliographic databases. In this context, we investigate a term extraction approach that automatically identifies expertise topics and builds expert profiles from scientific publications, providing an alternative measure of expertise.

In the NLP field, a range of applications can be used to provide information on the virus and the pandemic; machine translation can make that information available in many languages, text analytics can be exploited in identifying trends in society around issues resulting from the crisis, and knowledge extraction can help scientists in understanding connections between isolated studies.

The COVID-19 Open Research Dataset (CORD-19[1]), compiled and made available by the Allen Institute for AI, has enabled the application of a wide variety of NLP and other approaches to a very large dataset of COVID-19 related publications ([Wang et al., 2020](#)). Common ones as reported on the related Kaggle competition page[2] focus on: i) document clustering ; ii) question answering; iii) automatic creation of summary tables ; iv) biomedical knowledge graph construction; v) text analytics applications around the spread of the virus. In this work, we focus instead on the identification of experts in the CORD-19 dataset. Our approach aims at automatically extracting terms that represent expertise topics (term extraction step), and, by splitting the CORD-19 dataset into five sub-corpora corresponding to ERC (European Research Council[3]) review panels, ranking authors of papers that relate to these expertise topics (expert finding step). To evaluate our approach, we compare the automatically retrieved experts with established expertise (ERC panel members) in relevant subareas based on a popular impact metric, the *h-index* ([Hirsch, 2005](#)).

This paper is organised as follows: first we give an overview of related work in section [2](#), we present our approach in section [3](#), then we detail the set up of the experiments in section [4](#), and finally we present and discuss the results in section [5](#).

---

[1] https://www.semanticscholar.org/cord19
[2] https://www.kaggle.com/allen-institute-for-ai/CORD-19-research-challenge/kernels
[3] https://erc.europa.eu/

## 2 Related Work

Expert finding originated with different techniques in information retrieval, recommender systems, social network analysis and NLP (Maybury, 2006; Balog et al., 2012). Common approaches rank experts according to profile matching or voting (Fang and Zhai, 2007; Macdonald and Ounis, 2008). Profiles can be based on manually defined expert databases or knowledge maps (Davenport et al., 1998), profiles generated from social networks (Bekkerman and McCallum, 2005; Karimzadehgan et al., 2009), or through automatic expertise topic extraction from relevant unstructured textual data (Griffiths and Steyvers, 2004). Approaches based on voting represent expertise topics as queries and authors of documents as potential experts on those topics. Authors are ranked according to number of documents retrieved, i.e. each retrieved document constitutes an expertise vote for that author. Expert finding in the biomedical field has been explored in recent years by use of voting over terms from the Medical Subject Headings (MeSH) thesaurus (Singh et al., 2013) or through profile matching on the basis of language models (Wang et al., 2015). Our approach is based on Bordea (2013), which implements expert finding through voting over automatically extracted terms from text. In the context of this paper, we use the current implementation of this approach in the Saffron knowledge extraction framework[4].

## 3 Approach

In this section we give a brief overview of two automatic term extraction approaches and of the expert finding approach used for searching people instead of documents.

### 3.1 Index term ranking

Considering a subset of the publications is annotated with index terms from a controlled vocabulary, the question is how to rank these terms to identify broad research trends within a domain-specific corpus. A solution is to rank the most important terms for the COVID corpus dataset by using the *tf-idf* based approach described in Bordea et al. (2019). This approach takes into consideration the number of CORD-19 articles annotated with an index term, normalised by the overall number of publications annotated with the term.

---

[4] https://github.com/insight-centre/saffron

### 3.2 Term extraction

As mentioned above, we employ a term extraction approach as implemented in the Saffron knowledge extraction framework to identify relevant expertise concepts in the domain. Saffron implements a pipeline, comprising of candidate term selection, scoring, and ranking/filtering. In the first step, noun phrases are identified by use of lemmatization and Part-of-Speech patterns. Several scoring functions are available in Saffron, including frequency, distribution within the corpus, use of reference corpora, etc. They can be combined using a weighted scoring algorithm, and aggregates them together (Zhang et al., 2008). Here we use the default settings, which combine the functions *comboBasic*, *weirdness*, *totalTfIdf*, *cValue* and *residualIdf* (described in (Astrakhantsev, 2016)), with weight emphasis on the *comboBasic* function, based on Bordea (2013). It modifies *C-value* (Frantzi et al., 2000) by reinforcing the weight on terms embedded in or embedding other terms. This approach was shown to extract more specialized terms, which makes it very suitable for targeting expertise. Finally, the candidate terms are ranked by score and the top N terms are selected as the set of terms to be used in the expert finding step.

### 3.3 Expert Finding

In a second focus of this work, we employ term extraction to automate the identification of biomedical domain experts within their specialized expertise areas. Expert finding allows to efficiently locate individuals with a particular domain-specific skill or a proven expertise in a domain, and by extension help connect experts with each other and favor collaborations.

We are following the expert finding approach introduced by Bordea (2010, 2013) which measures the relevance of a term (i.e. an expertise topic), for an author (i.e. an expert). In this method, authors are ranked according to how frequently they mention each extracted term, noted $t$, which is calculated by means of *tf-irf* (Term Frequency-Inverse Researcher Frequency):

$$\text{tf-irf}_a(t) = \sum_{d \in D_a} \text{tf-idf}_d(t)$$

Where $D_a$ is the set of documents authored by author, $a$, and *tf-idf* is calculated to assess the

importance of the term in the corpus.

## 4 Experimental Setup

Our experiments are divided into two parts, corresponding to the term extraction and expert finding steps. Here, we consider extracted terms as expertise topics, against which experts (authors of the papers) are assessed and ranked. To evaluate our experiments, we compare our results to established resources, namely for term extraction the MeSH thesaurus, a controlled and hierarchically-organized vocabulary, and the widely used impact metric *h-index* for assessing expertise.

### 4.1 Resources

First, we present the different resources utilized for the accomplishment of this work.

#### 4.1.1 CORD-19 Dataset

The CORD-19 dataset comprises of COVID-19 and coronavirus related research articles from many different resources, including open archives such as PubMed[5], bioRxiv[6] and medRxiv[7]. The dataset is now updated daily, and we selected the release of 15/05/2020 for these experiments. This study is concerned with finding coronavirus related terms from the corpus. For the construction of the corpus on which we run the experiments, we selected the abstracts of the papers as they represent a complete summary of the content of the article and contain its key concepts. In addition to the abstracts, we collected the following provided metadata: the source of the paper, title, authors, date of publication, and for the papers sourced from PubMed, their corresponding PubMed identifiers (referred to as PMID in the rest of the paper). By focusing on this particular data content, we discarded all papers for which an abstract was not provided. Furthermore, we discarded from the final corpus all pre-prints from bioRxiv and MedRxiv, as a peer review process was deemed to us a necessary requirement to consider a scientific study approved in the field. We refer to this version of the CORD-19 dataset as used in this paper as the COVID corpus. In total, this corpus is made up of 96,167 paper abstracts, including 88,804 indexed with PMIDs and constitutes 21,713,354 tokens and 423,527 types.

#### 4.1.2 MeSH Thesaurus

We looked for all available articles in the MEDLINE bibliographic database, using the PubMed search engine and the provided PMIDs from the COVID corpus metadata. We collected (MeSH) index terms manually assigned to the articles in PubMed from the MeSH thesaurus. This resource provides valuable information about synonyms and term variants, and will be used to evaluate automatic extraction of terms.

Although very useful, MeSH is not without limitations because recently published papers are not instantly annotated with MeSH terms. Although almost 94% of the PubMed articles from the COVID corpus were accessible when we ran the analysis, only 56% were annotated with MeSH terms.

#### 4.1.3 ERC Review Panels

We make use of the taxonomy of scientific areas identified by the ERC to describe the expertise of evaluation panels[8] in order to split our main COVID corpus into relevant subareas. The following Life Sciences (LS) areas were considered to be the most relevant for the COVID context:

- LS1 Molecular Biology, Biochemistry, Structural Biology and Molecular Biophysics
- LS2 Genetics, 'Omics', Bioinformatics and Systems Biology
- LS3 Cellular and Developmental Biology
- LS6 Immunity and Infection
- LS7 Applied Medical Technologies, Diagnostics, Therapies and Public Health

Each area covers between 10 and 15 subareas, each accompanied by a list of panel descriptors (topics describing fields of research covered). Moreover, each ERC subarea can be linked to corresponding MeSH terms, and further be used to associate related publications from the COVID corpus to ERC subareas, using the PMIDs as described in section 4.1.2. Scientific articles may be assigned to multiple subareas at the same time, as different MeSH terms may be associated with different ERC areas.

### 4.2 Index Terms Experiments

A considerable amount of publications available through PubMed are manually annotated with index terms from the MeSH controlled vocabulary.

---

[5]https://pubmed.ncbi.nlm.nih.gov/
[6]bioRxiv: https://www.biorxiv.org/
[7]MedRxiv: https://www.medrxiv.org/

[8]https://erc.europa.eu/sites/default/files/document/file/ERC_Panel_structure_2020.pdf

In this experiment we aim to identify broad terms that can be used to identify expertise as described in section 3.1.

**Evaluation approach** The top ranked MeSH terms in the context of COVID-19 are manually analysed by an expert to identify relevant terms. For this purpose we limited our evaluation to the top 500 terms.

### 4.3 Term Extraction Experiments

We performed the term extraction experiments on the COVID corpus described in section 4.1.1.

Within the customizable features made available by Saffron for the term extraction step, we chose to generate terms of a minimum of two and a maximum of five words. This is motivated by the principle that longer words are more specific that shorter ones (Bordea, 2013) and identifying more specific concepts can potentially highlight more distinctive aspects or information than general ones. We selected the default settings provided by Saffron for the selection and ranking of terms, as described in section 3.2.

**Evaluation approach** The terms from the MeSH thesaurus are used as a gold standard to evaluate terms automatically extracted in the term extraction phase. We searched whether the 500 top terms extracted by the tool appeared in the MeSH terms, by using the MeSH Browser[9]. We consider both exact matches and partial matches in the evaluation, that is terms that are correct but that can also be a sub-string of the gold standard term.

### 4.4 Expert Finding Experiments

Since the research in coronavirus related diseases covers many sub-fields in the biomedical domain, and in order to identify more specialized expertise among researchers, we base our work on established classifications to perform the expert finding task on more specialized corpora, i.e. each of the ERC subareas of the COVID corpus split as described in section 4.1.3.

We used Saffron to perform our experiment, which extracts and then calculates for each term the *tf-irf* author score, as described in section 3.3. Since many extracted terms were shared across all of the ERC subareas, we asked a domain expert to discard the terms which were not specific to the

---

[9] https://meshb.nlm.nih.gov/search

| ERC subarea | Short name | Panel Members |
|---|---|---|
| LS1 | Molecular Biology | 180 |
| LS2 | Genetics | 181 |
| LS3 | Cell Biology | 170 |
| LS6 | Immunity&Infection | 157 |
| LS7 | Applied Tech. | 202 |

Table 1: ERC panel members per subarea from 2012

particular subarea. We used these new lists of terms to retrieve the experts.

**Evaluation approach** In this context, we consider a term as an area of expertise of an expert. Evaluating the competence and establishment of research experts in a domain is still an open question. It raises the issue of what and when to consider a person to be an expert, and the criteria vary between fields of research, e.g. the number of citations, the place of publication (main conferences, main journals in the domain), if the person was the first author of the paper (in some fields the author order is not taken into account). It is also hard to take into account the specific subareas of expertise of a person, especially through time, as opposed to a more global field of expertise.

Due to a lack of resources and standards to estimate the correctness of authors' expertise, for the evaluation we instead focused on assessing whether authors were recognised among the research community (impact). We propose here two angles for evaluation.

**ERC panel members evaluation** Since the expert finding experiment is carried out on the COVID corpus split into ERC subareas, we decided to retrieve ERC subareas panel members to create a gold standard of experts. Panel members are indeed established researchers chosen for their evidence-based expertise in the domain, which allows us to reasonably use their names as a base for evaluation. We retrieved the ERC panel members as available from their website[10] since 2012 (see distribution of members per ERC subarea in Table 1). We then searched the first 10 experts in the panel members list for each ERC subarea, evaluating in this way the precision at 10.

**Metrics-based evaluation** For the second evaluation approach, we looked at common author-level expert metrics in the research community.

---

[10] https://erc.europa.eu/document-category/evaluation-panels

The h-index is widely used by the scientific community. It takes into account the amount of publications and citations and is more an estimation of the impact of an author in the domain than a measure of expertise. It is calculated in publication platforms such as Google Scholar. In this evaluation, we manually retrieved the *h-index* score for the top three experts of the top three terms extracted by Saffron for each ERC subarea. This allows us to analyze how our calculations for expert finding relates to this popular way of measuring impact. When available, we preferred Google Scholar as the source to obtain the *h-index* as it showed to be the most complete resource, and selected the highest score available between Scopus, Semantic Scholar and Mendeley when not available. We note that *h-index* is a set metric, but the threshold above which a person is considered as an impactful expert will vary from field to field. In order to have a better idea of an "good enough" *h-index* to consider an author as expert, we randomly selected 20 panel members for each ERC subarea and retrieved their *h-index* to use as a point of reference to compare with.

## 5 Results

Here we discuss the results for our experiments on term extraction and expert finding.

### 5.1 Index Term Ranking

Highly ranked MeSH terms include: *Coronavirus Infections*, *Severe Acute Respiratory Syndrome*, *Human Influenza*, *Viral RNA*, and *Respiratory Tract Infections*. While the first two terms were used to retrieve the initial CORD-19 corpus, the last three can be considered to extend the search query to other relevant terms. Several highly ranked terms are closely related in the MeSH hierarchy, therefore the top 250 terms are further analysed to identify prominent MeSH subtrees. These include: *Viruses [B04]*, *Infections [C01]*, *Investigative Techniques [E05]*, *Genetic Phenomena [G05]*, and *Environment and Public Health [N06]*.

### 5.2 Term Extraction

Table 2 shows a sample of 10 extracted terms obtained from the term extraction experiment on the whole COVID corpus, representing some prominent corpus terms with identified novel terms not present in the MeSH thesaurus.

We observed in the results the presence of expected coronavirus related terms, such as *acute*

| Sample terms | Novel terms |
|---|---|
| polymerase chain reaction | TGEV infection |
| respiratory infection | coronavirus pneumonia |
| epithelial cell | LAMP assay |
| control group | protective immunity |
| clinical trial | risk perception |
| syncytial virus | social distancing |
| spike protein | vaccine development |
| lymph node | virus detection |
| antiviral activity | novel coronavirus |
| mechanical ventilation | care worker |

Table 2: Sample of 10 extracted terms and terms that are not yet described in MeSH

*respiratory*, *severe acute respiratory*, *severe acute respiratory syndrome*. Each of them is extracted as a separate valid entity, as they appeared in the corpus as part and within other multi-word expressions, e.g. *severe acute respiratory infection, hypoxemic acute respiratory failure, acute respiratory syndrome*.

In total, 33% of the top 500 Saffron terms extracted for the COVID corpus can be found in the MeSH thesaurus using exact match search (e.g., *innate immunity*, *heart failure*), while 18.8% terms can be found with a partial overlap (e.g., *fatality rate* vs. *case fatality rate*). In addition to these, 34.6% were found to be correct terms but not currently covered by MeSH, including *coronavirus pneumonia* and *TGEV infection*, an acronym for *Transmissible gastroenteritis coronavirus*. This approach could therefore be beneficial in helping extend the MeSH thesaurus and better cover this active research field by identifying novel terms. By combining the above scores together, we evaluate the overall precision for term extraction at **86.4%**.

An error analysis of the incorrectly extracted terms showed that there are sometimes issues with identifying concept boundaries (e.g., *east respiratory* vs. *middle east respiratory syndrome*. Terms separated by a hyphen are also more difficult to extract (e.g., *enzyme-linked immunosorbent assay*). Also, frequently occurring expressions are occasionally extracted as terms, such as *first time*, *wide range*, *large number*, and *recent year*.

### 5.3 Expert Finding

Table 4 presents a sample of results from the expert finding extraction. It shows the top 10 extracted terms from the COVID corpus for the *Immunity-Infection* ERC subarea, associated with identified (top ranked) experts.

| LS1 | LS2 | LS3 | LS6 | LS7 |
|---|---|---|---|---|
| structural protein | gene expression | cell line | viral infection | infection control |
| fusion protein | phylogenetic analysis | epithelial cell | viral load | global health |
| crystal structure | viral genome | dendritic cell | antibody response | surveillance system |
| recombinant protein | genome sequence | cell fusion | innate immune response | disease outbreak |
| sialic acid | rna synthesis | cell cycle | antiviral drug | emergency department |

Table 3: Selected Saffron terms per ERC subarea, where *LS1* stands for *Molecular Biology*, *LS2* for *Genetics*, *LS3* for *Cell Biology*, *LS6* for *Immunity&Infection*, *LS7* for *Applied Tech.*

| Saffron Term | Top Ranked Expert |
|---|---|
| viral infection | Hershenson, Marc B. |
| viral load | Chan, Kwok-Hung |
| antibody response | Jiang, Shibo |
| innate immune response | Ito, Yoko |
| antiviral drug | Peng, Guiqing |
| inflammatory response | Zhou, Yusen |
| cytopathic effect | Mizutani, Tetsuya |
| acute respiratory infection | Barrett, Bruce |
| bacterial infection | Morozumi, Miyuki |
| antiviral response | Jin, Dong-Yan |

Table 4: Top ranked experts for 10 Saffron terms from the *Immunity Infection* ERC subarea

| Area | Short name | Percentage |
|---|---|---|
| LS1 | Molecular biology | 89% |
| LS2 | Genetics | 56% |
| LS3 | Cell Biology | 100% |
| LS6 | Immunity&Infection | 89% |
| LS7 | Applied tech. | 67% |

Table 5: Number of experts extracted above the minimum *h-index* among panel members

**Evaluation using ERC panel members** Across all subareas, only one expert identified using our approach was found to match an ERC panel member: *Volker Thiel*. Several interpretations can be drawn to explain this result. We observed that many of the extracted experts originated from Asia, among which specifically several authors from the Department of Microbiology in the Hong Kong University. The ERC panel members on the contrary contained only a few Asian specialists. One could interpret this as the panels being rather centred on Europe and North America. As a second explanation, one can argue that the Asian research community is currently having a head start on this particular topic, given the origin and natural timeline of the epidemic.

**Evaluation using impact metrics** As mentioned in section 4.4, we selected the top three Saffron terms extracted for each subarea and corresponding top three identified experts to perform the manual evaluation by retrieving their *h-index*.

Table 5 shows the percentage of experts in our evaluation that are above the minimum *h-index* threshold set by the 20 panel members who constitute our gold standard for each subarea. We can see that in general a majority of identified experts are above the threshold. Another way of comparing these values is given through a boxplot diagram in Figures 1 (all subareas combined) and 2 (split per subarea, *LS#* representing the gold standard for

the subareas, and *S_LS#* the experts identified using Saffron). Figure 1 shows that in general the *h-index* range for Saffron-identified experts is displaced towards lower values, which is to be expected given that we are focusing on COVID-19 experts while ERC experts can have a much broader expertise. Nonetheless, the median values for *h-index* in both cases are around an *h-index* of 60, with maximum values of *h-index* at 120.

Taking a closer look at the detailed results by subarea in Figure 2, we notice that the median values are actually higher for Saffron experts in Immunity & Infection and Applied Medical Technologies. These are both areas that are highly connected with the COVID-19 sanitary crisis, which is mainly a Public Health issue. The Saffron-identified experts have lower median values for Molecular Biology, Genetics and Cell Biology, with greater disparities between the *h-index* for the first two subareas. This might indicate higher traction for applied research compared to the more theoretical research areas, which is reasonable in the immediate aftermath of a crisis.

# 6 Discussion

We note that we are missing a certain amount of information regarding the newest research since 2020 papers are not yet indexed in PudMed. We can not therefore consider our analysis complete, but it can already give a good preliminary result of the trend. We identify some benefits in using a different approach to expert finding through text mining, as opposed to widely used metrics which revealed some limitations.

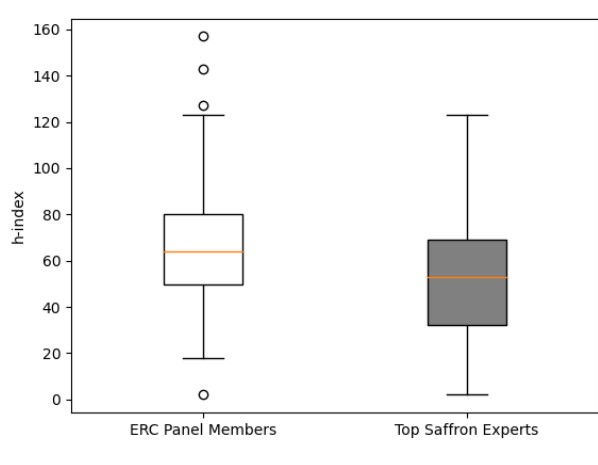

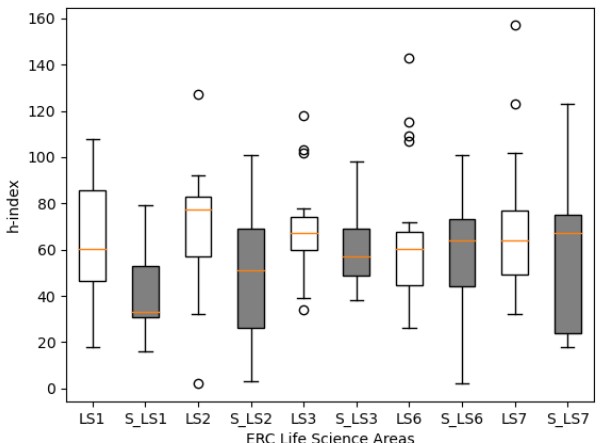

Figure 1: Comparison of *h-index* for ERC panel members with Covid-19 experts identified by our approach (in gray)

Figure 2: Comparison of *h-index* for ERC panel members with Covid-19 experts identified by our approach (in gray) by ERC area

Firstly, the *h-index* can vary quite significantly in between the platforms, directly depending on how complete the bibliography of an author on a particular publications platform is. Numbers can vary greatly, and Google scholar has shown to give higher *h-index* (therefore is a more complete source) as opposed to Scopus or Semantic Scholar, when all three sources were available. Author consolidation is also another problem, as papers can be attributed to a particular expert profile while not corresponding to the right author. It is left to the author concerned to verify the correctness of the sources attributed to him, and even then the procedure to claim or discard a publication is not straightforward.

Secondly, not all experts have a profile on platforms like Google Scholar, which does not mean they are not experts. Some researchers have a Wikipedia page but no Google Scholar profile (eg. *Kwok-Hung Chan*, expert in Genetics or *Zihe Rao*, expert in the Molecular Biology field). Not all "mainstream" channels are used by all researchers, and this disparity is a problem to link them together. Communities in different areas of research may have a different culture in terms of outreach and demonstration of their work. Identifying experts and their areas of expertise from the text used in publications themselves can be a way to bridge this gap and put the light on less "popular" researchers but experienced all the same. This method would therefore allow for a better inclusiveness.

Thirdly, common and established resources used

to classify papers according to their area of research have also their limitations. It takes some time to index recent papers with domain-specific resources, as we showed with PubMed. Furthermore, domains of research are constantly evolving. As we showed in our evaluation of terms against the MeSh thesaurus, the resource was lacking some important new concepts which emerged very recently. Such resources may take some time to update as deep efforts in consultation with domain experts are necessary. The often rigid structure of the classification may also not allow an easy extension and adaptation to new sub-fields. They are often also very complex. As an example, the Unified Medical Language System® (UMLS®)[11] includes many complex and interlinked lexical resources. A deep expertise both in the domain and in the specific resource(s) is then necessary for any classification or expert identification. With our approach based on text mining, we show how we can easily identify established and new sub-fields without the need for external domain-specific resources,and automatically link them to their corresponding experts, without the need to wait for the field specialist to update to the current state of the research.

This all shows the potential and the need of a data-driven method to connect experts together who do not know about each other, diminishing biased created by closed networks, opening the community in these areas of expertise, and discover new

---

[11] https://www.nlm.nih.gov/research/umls/index.html

areas of expertise. Saffron expert finding, being data-driven and not field-tied can allow in this way for a more independent interpretation of expertise.

## 7 Conclusion

We carried out an exploration of the CORD-19 dataset through the lens of expert finding. A preliminary analysis based on a reduced subset of extracted terms and experts show promising results. We showed that term extraction can take into account more recent research not yet covered by established domain-specific resources, and can even help the community by drawing attention to novel terms. The proposed expert finding approach has the potential to provide better visibility either to new emerging communities of experts, or to communities that are not yet part of the mainstream research path. A benefit of using a data-driven method, is to be able to identify experts purely based on their published work.

Limited access to large-scale resources for expert evaluation is a barrier for a more robust assessment of the impact that this approach could have on the domain. As mentioned, *h-index* values vary considerably between resources, involving tedious work for manual retrieval. The Allen Institute for AI is currently considering linking bibliographic entries to the corresponding papers in Semantic Scholar. With this information available from the corpus itself, a more thorough research of *h-index* for all the extracted experts could automatically be performed, allowing us to map our results to Semantic Scholar. However as pointed out in the discussion, the *h-index* has its own limitations, therefore further research is still needed on other acceptable metrics and methods for expertise evaluation.

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
