# OpenReview forum: "A Term Extraction Approach to Expert Finding on the COVID-19 Open Research Dataset"
_EMNLP/2020/Workshop/NLP-COVID — Submitted to NLP-COVID19-EMNLP_

### Official Review · AnonReviewer3 · 2020-09-24
**Needs a clearer/improved evaluation**

**Rating:** 5
**Confidence:** 2

**Review:**

The authors combine an off-the-shelf term extraction system (Saffron) and manually collected MeSH terms with a tf-idf based method (Bordea (2010, 2013)) for identifying scientific experts in under-indexed literature in the CORD-19 dataset.


Reasons to accept:
- the authors apply existing technologies to a new problem


Reasons to reject:
- I have trouble following the evaluation; it has many suspect steps and I feel that they are underaddressed. I would like a proper metric here instead of a series of ad-hoc approaches. As an example h-index is not a good measure for expertise in a novel and/or developing area, which is one of the main focuses of the paper.
- The authors present a pipelined approach for this technique, but they don't investigate how errors propagate within the pipeline. I don't have a sense for what their numbers mean. What happens if we strictly use MeSH terms and manually tagged documents? How does this impact the overall expert identification? How sensitive is the expert identification scoring to the h-index; what's the distribution of scores?
- this is a straightforward application of existing technology, not a methods development (this is a much weaker complaint than the evaluation angle)
- some of the methodology is arbitrary but would likely matter in practice, e.g. choosing how many extracted terms to use?
- the subarea splits and use of ERC experts seem to mostly distract given that there seem to be few extracted ERC experts, rendering both the ERC and the split moot.

Questions:
- what instructions was the expert annotator given for the MeSH terms?
- low quality term extraction may lead to overly aggressive MeSH term matching. How did you compensate for this?
- what proportion of researchers are ERC panel members? This is a crucial point to understanding the evaluation of expert identification
- what was the h-index threshold you settled on?
- why use this word-mesh-term overlap instead of an existing MeSH tagger?

I am deeply dissatisfied with the evaluation methodology. I would be much happier if a less ad-hoc version were performed on a limited subset. Given the lack of results using ERC panels as experts, a large portion of the paper seems irrelevant (null results are fine, but there's no meaningful investigation of the null result). I think that the idea of expert discovery could be explored on better tagged datasets (e.g. the portion of CORD-19 that has already been indexed, or any other subset of PubMed), allowing for easier measurement of the decay for any portion of the pipeline.

---

### Official Review · AnonReviewer1 · 2020-09-25
**A term extraction method but the evaluation is of concern**

**Rating:** 5
**Confidence:** 3

**Review:**

This study proposes a term extraction method to extract expertise topics from coronavirus related papers which aims to find experts accordingly.  My main concern is on the evaluation part.

First, the study does not compare with any baseline methods or other term extraction methods. The effectiveness of the proposed method is thus not clear.

Second, it is confusing that using MeSH terms for evaluating extracted expertise topics. MeSH terms are not designed for expertise topics at all. It annotates the terms wherever applicable to the ontology, not key terms or expertise terms. For instance, https://pubmed.ncbi.nlm.nih.gov/32134205/ has a MeSH term 'Humans', which is not the main term of the study at all. Having an overlap between MeSH terms does not necessarily imply the expertise topics are extracted.

Third, it is even more puzzled on using h-index to 'quantify' the expertise... Is it a reasonable metric? Why an h-index of 20 was selected?

Fourth, looking at the extracted terms, it seems that they are not COVID-19 specific. For example, 'infection control'  and 'antibody response' are general terms for different types of coronaviruses. Whether it can find specific expertise terms for COVID-19 is not clear. The study could apply the method to LitCovid (https://www.ncbi.nlm.nih.gov/research/coronavirus/) which consists of PubMed articles that are on COVID-19 and it has curated topics to conduct a case study for evaluation. For instance, compare the extracted terms from COVID-19 articles with the terms from articles on other coronaviruses.

---

### Official Review · AnonReviewer2 · 2020-09-25
**methods used are reasonable but the evaluation is problematic**

**Rating:** 4
**Confidence:** 3

**Review:**

The paper explores the task of experts extraction and expertise topic extraction from COVID-related literature. While the methods seem reasonable, the evaluation is quite problematic, as explained below:

* For term extraction, the evaluation is mainly based on the MeSH terms, which I don't think is appropriate. Based on "When labelling an article, indexers select terms only from the official MeSH list – never other spellings or variations" ([quote from here](https://onlinelibrary.wiley.com/doi/full/10.1111/ijcp.12767#:~:text=MeSH%20terms%20are%20official%20words,never%20other%20spellings%20or%20variations)), the evaluation using MeSH terms risks penalizing words extracted that share the same topic but different surface forms with the MeSH terms. For example, "heart attack" would be not be recognized as a successful extraction since the word used in the list is "Myocardial Infarction".

* For expert extraction, there are two ways as evaluation:  ERC panel member matching and h-index. Both evaluation methods are quite problematic.
    *   For ERC panel members matching, the authors have already pointed out its limitation in the discussion "The ERC panel members on the contrary contained only a few Asian specialists ...  the panels being rather centred on Europe and North America". So the evaluation is biased and tells little about the results.
    * The h-index tells mainly the productivity and citation impact of an author. It is just not an appropriate measure for expertise. For example, a person has a higher h-index but few work related to COVID-19 would potentially be recognized as a top expert by this metric. Furthermore, the authors claimed
> "Typically, researchers rely on metrics of publication and citation impact to evaluate the level of expertise of their peers, but these metrics are not as reliable nor directly comparable across scientific fields and bibliographic databases".

    But I don't see any advantage of using h-index over other metrics of publication and citation impact.  Please clarify this.
* There is no comparison between the proposed method and any other baseline. So it is hard to interpret all the reported results.


Suggestion:
As an NLPer, I had a hard time finding the definition of MeSH terms and the annotation criteria for MeSH terms. It would be better if the authors could add some definition and details about MeSH in the paper.